# Development and Verification of Prototypical Office Buildings Models Using the National Building Energy Consumption Survey in Korea

**Hye-Jin Kim** [1] , **Do-Young Choi** [2] **and Donghyun Seo** [1,*]

1   Department of Architectural Engineering, Chungbuk National University, 1 Chungdae-ro, Seowon-gu, Cheongju, Chungbuk 28644, Korea; kimhj1755@chungbuk.ac.kr
2   Korea Energy Economics Institute, 405-11, Jongga-ro, Jung-gu, Ulsan 44543, Korea; dychoi@keei.re.kr
*   Correspondence: sirdh1@g.cbnu.ac.kr

**Abstract:** In the early 2000s, the Korean government mandated the construction of only zero-energy residential buildings by 2025 and for non-residential buildings from 2030. Two decades since the start of building energy policy enforcement, Korean experts believe that it is time to evaluate its impact. However, few studies have systematically and extensively examined the energy consumption characteristics of the non-residential building stock. In this study, a framework development is implemented for defining non-residential prototypical office buildings based on Korea's first large-scale non-residential building survey result from the Korea Energy Economics Institute (KEEI). Then, a detailed building energy model of the defined prototypical building is constructed to verify the model's energy estimation against observed energy consumption. As an application of the model, a case study for energy policy evaluation utilizing the constructed prototypical building model is presented. Every researcher and county may have their own circumstances when gathering definition data. However, by using the best available representative data, this suggested framework may result in informed decisions regarding energy policy development and evaluation. In addition, the mitigation of greenhouse gases from buildings may be expedited.

**Keywords:** non-residential building; prototypical building; reference building; building energy model; end-use energy consumption





## 1. Introduction

The prototypical building (PB) concept is frequently used to estimate building energy consumption during energy policy formulation, as well as technology development and evaluation processes. Because a PB represents typical energy consumption characteristics (i.e., monthly electricity and fuel consumption, end-use of a particular building stock), the PB energy model results represent the entire building stock. PBs are widely used by policymakers, researchers, architects, engineers, and other stakeholders to establish long-term energy policies by evaluating the building energy savings before and after policy application [1,2]. Reference building (a different term for PB) models of US-DOE are used for policy evaluation by defining PB energy models for building stocks that represent 70% of the gross area of the non-residential building stock. A few research reports [3,4] from Pacific Northwest National Laboratory present the mechanisms by which policies have contributed to energy efficiency enhancement in the non-residential building sector. Numerous researchers and engineers have used PBs in their energy models to verify the energy performance of the building elements they designed, such as windows, shades, lighting, and heating, ventilation, and air conditioning (HVAC) systems because PBs contribute to the rapid progress and objectivity of their research [5]. A widely-used guide for this purpose is the ASHRAE Standard 90.1 (App G), which provides detailed building energy model procedures and methods based on prototypical non-residential building

data. Most "Leadership in Energy and Environmental Design (LEED)" certification projects follow this guide to obtain LEED credits.

A vast amount of data is required to define the characteristics of PBs, such as the shape of the building, thermal transmittance of envelopes, heating and cooling equipment type, and operational information in addition to building energy consumption. For example, [6] defined various U.S. PB models based on the 2003 Commercial Buildings Energy Consumption Survey (CBECS, EIA, 2005) data, ASHRAE 90.1, and published research results.

In the early 2000s, the Korean government mandated the construction of only zero-energy residential buildings by 2025; for non-residential buildings, the mandate starts from the year 2030. To this end, the thermal transmittance, also known as the U-value of building envelopes, has been enforced by the government since 2001 (Figure 1). After two decades of enforcement, Korean experts are starting to evaluate the impact of the enforced building code. However, few studies have systematically and extensively examined the energy consumption characteristics of the residential and non-residential building stock. Using PBs is the most cost-efficient method to evaluate the building performance for a variety of building stocks under different conditions. However, the development of reliable PBs in Korea is still relatively immature owing to the scarcity of energy-related building data and published research papers. Recently, the Korea Energy Economics Institute (KEEI) has carried out an energy survey of 1000 non-residential buildings in the Seoul Metropolitan Area to characterize the energy consumption features of non-residential buildings [2]. The survey enabled the development of PBs with statistically processed data on building energy consumption.

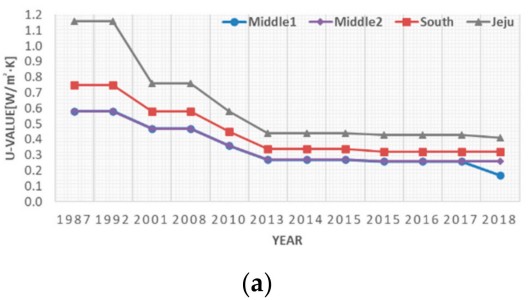

(**a**)

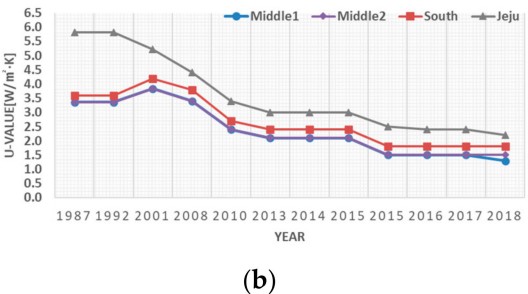

(**b**)

**Figure 1.** Enhancement of the thermal transmittance exterior walls and windows since 1987: (**a**) Exterior walls; (**b**) Exterior windows.

In this study, a framework for defining non-residential PBs was developed based on the survey results from the KEEI. Subsequently, detailed building energy models of the defined PBs were constructed to simulate and verify energy consumption. Section 2 provides a review of previous research on non-residential PBs. Section 3 describes the features of the KEEI survey including questionnaire development and sampling and quality control of survey data. Section 4 presents the development process of the PB definition framework. Section 5 demonstrates the process starting from the definition of PB data and verification of the PB energy model to the application of the model for energy policy evaluation.

## 2. Review of Previous Research and Definition of Non-Residential Prototypical Buildings

There is no commonly accepted definition for PBs. Reference [7] defined PBs as "a DOE-2 input model that is a representative of the average building stock for a particular building type." Reference [8] defined PBs as "buildings characterized by their functionality and geographic location, including indoor and outdoor climate conditions." Reference [9] defined a PB as "a virtual building made with various elements defining typical architec-

tural, mechanical, and operational characteristics and energy performance of a building stock." In this study, the PB definition process conforms to the principles presented in [9].

The items that correspond to PBs are well documented in [6] and classified into four major categories, namely "Program," "Form," "Fabric," and "Equipment." Each category is composed of subcategories. In this study, category names were modified to be more intuitive, and the "Energy consumption and generation" category was added (Table 1).

**Table 1.** Five categories and the exemplar components of each for prototypical building (PB) definition

| General Information | Architectural Features | Building Systems | Operation | Energy Consumption and Generation |
|---|---|---|---|---|
| • Region<br>• Building type<br>• Shape<br>• Number of floors<br>• Floor height<br>• Gross area<br>• Building area<br>• Conditioned area | • Envelope<br>• U-value<br>• Window-to-wall ratio (WWR)<br>• Window type<br>• Shading<br>• Location of a parking lot | • Lighting<br>• Office equipment<br>• Plants<br>• HVAC system<br>• Domestic hot water system<br>• Renewable energy system | • Building operation<br>• Occupancy<br>• System operation<br>  - Heating<br>  - Cooling<br>  - Ventilation<br>  - Domestic hot water etc.<br>• Heating/cooling setpoint | Monthly:<br>• Electricity<br>• Fuel<br>• Renewable energy |

In Table 1, "General information" is a category that incorporates building shape and geographical location. "Architectural features" include data for calculating the building heating and cooling loads and variables for the physical properties of each building construction material. The "Building systems" category provides information regarding the mechanical and electrical equipment of buildings that directly consume energy. The "Operation" category contains variables related to occupant's behavior and the operation schedule of equipment that affects the energy consumption of the "Building systems." "Energy consumption and generation" is necessary to define the typical monthly energy consumption and production in the building. Energy sources in office buildings can be divided into electricity and fuel, wherein fuel includes all types of fossil fuels. Renewable energy is also included in this category. The value for each energy source is used to verify the PB energy model simulation results. Note that the variables of each category in Table 1 are representative components and not the complete list of variables because the necessary variables vary depending on building type.

To assign reasonable values to these components, a vast amount of data is required. For example, the gross area range of medium-sized buildings in a population should be determined to calculate the gross area of a typical, medium-sized office building. Subsequently, the average or median value can be derived from the building group data. National building registration databases managed by governmental agencies are the most ideal. However, depending on the country, this information may not be available, and/or collecting this information can be time-consuming and expensive. For this reason, the gross area is calculated by examining the buildings sampled through robust statistical procedures or by collecting arbitrary building data from limited areas and target buildings. Therefore, the quality of data used for the definition process is important, and the representativeness of the defined PB can be determined by the type and quality of the datasets.

Data that can be used to define a PB can be collected from various sources, each with a different level of difficulty for obtaining information. For example, the insulation level of a building envelope, which may not be easily calculated without the building blueprints, can be estimated relatively easily using the building code and the building's year of completion. However, to calculate the typical gross floor area of specific building stock, it is necessary to follow a complex process of collecting and analyzing building registration data managed

by administrative agencies. In addition, different types of datasets can be used in research and survey reports on the characteristics of building energy consumption, statistical data on buildings, survey data on building use, and sales volumes of heating and cooling equipment.

It is nearly impossible to collect representative data uniformly to determine the values of the components listed in Table 1 because of the time and cost requirements of collecting and processing data. For example, the gross floor area of all sample buildings is available in the building management document, but because the air conditioning area is not of interest in general buildings, this variable depends on the manager's intuition or estimation rather than an accurate literature search. Therefore, most studies that have attempted to define a PB use a variety of datasets depending on the data availability for each definition component. As such, many types of data are required to define a PB, and the representativeness of the PB is determined by the quality of the data.

Reference [8] categorized the types of PBs ("Example Reference Building," "Real Reference Building," and "Theoretical Reference Building") based on data availability and definition methods. However, the "Example Reference Building" classified in this method is not a PB in a strict sense, rather a base-case building used in research targeting a specific building. There are few actual studies on the "Real Reference Building" definition, and the utilization of the results is quite limited. Reference [8] presented a "Theoretical Reference Building," which is generally used for research, and [10] conducted a study to subdivide the "Theoretical Reference Building" according to the difficulty and representativeness of the data collected.

In this study, a more detailed and systematic revised version of the classification criteria of [10] is presented: First, the level of useful data was divided into three stages, and the resulting PB was divided into three levels. The level of data was selected as the main criterion for the difficulty of collection and the comprehensiveness and representativeness of the data.

The lowest level (Level 0) indicates that data are relatively easy to obtain, and statistical representation is weak. Therefore, the value of the definition components can be determined based on a priori knowledge, engineering judgment, and design criteria, and thus Level 0 refers to "Empirical Data." Building energy codes, building codes, and technical handbooks applied to most building designs are examples of "Empirical Data." The next level (Level 1), "Limited Data," indicates that additional time and effort are required for the acquisition and analysis of the data. The number of buildings to be surveyed and the representativeness of the survey data are limited. These data are usually obtained from surveys and measurements conducted with targets restricted to a specific area or a building group. The highest level (Level 2) indicates that data are collected through large-scale surveys by various organizations and related to architecture, heating and cooling systems, lighting, and office equipment. These data are referred to as "Comprehensive Data," which include registered building databases from governmental agencies, data on demographics and living characteristics from the National Statistical Office, statistics on the energy supply of utilities, and nationwide sales data of aggregated HVAC systems.

Figure 2 shows the methodology used to create three different levels of PBs. Differentiation depends on the way different data are used. An empirical decision-based prototypical building (EPB) is defined by the expert's experience and engineering judgment. Rather than statistical data, indirect data that can be used as a basis for one's own experience or engineering judgment are used to determine the typical building size, insulation thickness, and class of HVAC equipment. EPB approximates the baseline building used in simple research because it lacks objectivity and has poor representativeness. References [11–14] used EPB building energy models to analyze the level of energy savings when energy-saving techniques are applied to baseline buildings.

A limited information-based prototypical building (LPB) is defined using statistical data with expert experience and assumptions. Although building information is collected to define a PB for a certain building stock, it is difficult to achieve representativeness

because statistical data are produced by arbitrarily selected samples without following a statistical sampling procedure. These data are usually collected through questionnaires, building site surveys, and manager/occupant interviews for limited sample buildings. Even if the data are based on a large-scale survey, the data used to define a PB only represent a specific part of the temporal, spatial, and survey items. References [9,15–17] used LPB-type building energy models.

Statistical analysis based on a prototypical building (SPB) uses statistical data regularly published by the state and public institutions for definition. The building data sampled and surveyed using statistical techniques are used to determine the value of each component defining the PB. For some values that are difficult to collect, the results for the EPBs and LPBs can be used. SBP is the most reliable method for defining PBs because the collected data are derived from systematic and extensive building survey results. References [6,18–21] used this SPB-type building energy model.

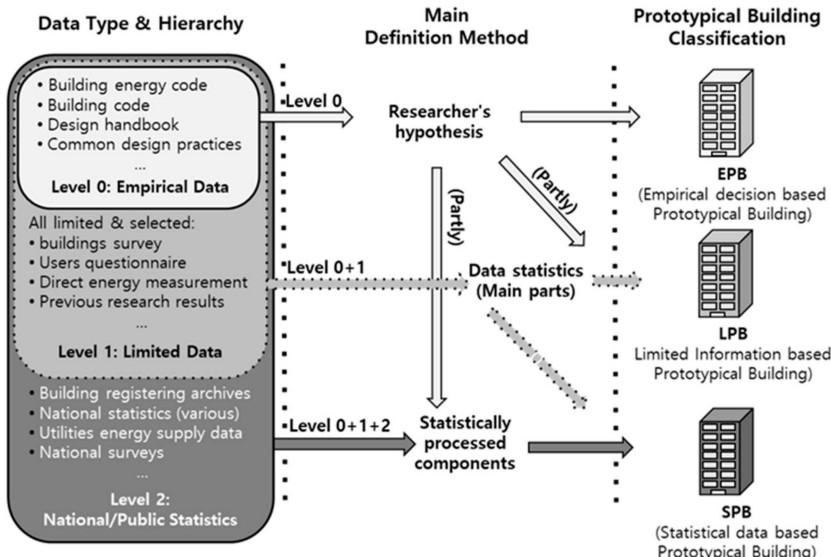

**Figure 2.** Classification of prototypical building definition with respect to data and methods.

## 3. Non-Residential Building Energy Consumption Survey by KEEI

To improve the understanding of energy consumption characteristics of non-residential buildings, which account for 48.8% of building energy consumption in Korea as of 2017 [22], the KEEI conducted a non-residential building energy consumption survey from 2014 to 2016 [6]. The purpose was to establish policies and technical strategies for reducing greenhouse gas emissions in the building sector.

The sample population for extracting buildings was limited to the commercial, medical, educational, office, telecommunication, and lodging sector in the Seoul Metropolitan Area. The 2013 National Building Registration Database was used for this purpose. Sampling was performed using the stratification extraction method, and the stratification variables include region (city and province), building use, and gross area. The target relative standard error of the annual energy consumption for each building type was used for the distribution of the extracted samples. The Neiman allocation method (reflecting the standard deviation) was used to allocate the number of samples per building type, region, and gross area. Throughout this process, 1000 sample buildings were extracted, but only 601 samples were used because of the high survey rejection and non-response ratio. The sampled buildings were classified into five gross area types (Mid-1: 3000–6000 m$^2$, Mid-2: 6000–10,000 m$^2$, Large-1: 10,000–30,000 m$^2$, Large-2: 30,000–50,000 m$^2$, Large-3: 50,000 m$^2$ or greater) to reflect the building energy consumption characteristics with respect to building size.

A principal objective of this survey was to identify the energy consumption of non-residential buildings by end-use (cooling, heating, hot water supply, lighting). Because most buildings do not have a system capable of measuring end-use consumption, it

is impossible to derive such results only based on the survey. To achieve the end-use goal, while increasing the reliability of responses, the survey questions were improved based on the following principles: 1. compose questions that do not require technical building knowledge; 2. add survey items that need to be input to building energy models such that one may reasonably estimate the end-use. Accordingly, additional questions were constructed such that architectural engineering data (e.g., thermal transmittance of envelopes, window thermal and spectral performance) and building operational data (i.e., occupancy, lighting, HVAC control schedules) may be directly or indirectly collected.

Table 2 summarizes the final questionnaire used to determine the value of each component in Table 1. The questionnaire consisted of questions on buildings, HVAC systems, lighting along with equipment energy use, and operational patterns. Table 3 summarizes the components necessary for the actual PB definition from the questionnaires and provides a comparison of items necessary for the definition (Definition) and items contained in the questionnaire (Survey).

To analyze the energy consumption characteristics of non-residential buildings, including estimation of end-use consumption in Korea, the first statistically systematic sampling accompanied by a questionnaire was implemented in this study. The results of this survey can be used in various disciplines. Methods for determining the PB input values and their utilization in the building energy model for evaluation of the energy policy are presented in Sections 4 and 5.

**Table 2.** Content summary of the overall questionnaire on non-residential building survey.

| I. Building | II. Energy Facilities | III. 2015 Energy Consumption |
|---|---|---|
| 1. General information | | |
| 1-1. Number of buildings | 1. Heating and cooling system | |
| 1-2. Typical floor shape | type and usage time | |
| 1-3. Floor height and ceiling height | 2. Renewable system type | |
| 1-4. Building orientation | 3. Air handling unit system type | |
| 1-5. Net area and condition area ratio | and usage time | 1. Monthly electric energy consumption |
| 1-6. Number of elevators and escalators | 3-1. Heating/cooling setpoint | 2. Monthly city gas energy consumption |
| 1-7. Vintage of building | temperature | 3. Monthly district energy consumption |
| 1-8. Energy efficiency rating | 3-2. Heating/cooling | 4. Monthly fuel energy consumption |
| 1-9. Number of floor and usage of each floor | maintenance temperature | |
| 2. Parking lot type and area | 3-3. Economizer system type | |
| 3. General schedule of a building | 4. Lighting power consumption | |
| 4. Types of windows and glass | and usage time | |
| 5. Type of exterior shade | 5. Electronic device type | |
| 6. Structure and shape of exterior walls and roofs | | |

**Table 3.** Comparison of components for actual PB definition and questionnaire.

| General information | **Definition** | Region | Building type | Shape | Number of floors | Floor height | Gross area | Building area | Conditioned zone area |
|---|---|---|---|---|---|---|---|---|---|
| | **Survey** | Region | Building type | • Typical floor<br>• Building direction | Number of floors | Floor height | Gross area | Building area | Conditioned zone area |
| **Architectural features** | **Definition** | Envelope | U-value | | WWR | Window type | Shading | | Location of a parking lot |
| | **Survey** | Roof &walls<br>• Structure<br>• Material | • Shape of roof<br>• The lowest floor surface<br>• Year of completion | | WWR | Glass type | • Window frame<br>• Shading | | • Location of a parking lot<br>• Parking lot area |
| **Building systems** | **Definition** | Lighting | Appliances | Plants | Domestic hot water | HVAC system | | Renewable energy system | |
| | **Survey** | • Power consumption<br>• Usage time | Type | System type * | Capacity | • Number of systems<br>• Pump capacity<br>• Purpose of use | | • Type<br>• Capacity<br>• Purpose of use | |
| **Operation** | **Definition** | Occupancy | | | System operating (heating/cooling, domestic hot water) | | | Heating/cooling setpoint | |
| | **Survey** | • Fixed/unfixed number of people<br>• Week/weekend average working hours per day | | | • Daily/annual average hours of use<br>• Continuous/non-continuous operation time | | | Heating/cooling setpoint | |
| **Energy consumption** | **Definition** | Monthly Electricity | | | Monthly Fuels | | | Monthly Renewable Energy | |
| | **Survey** | • Electricity power<br>• Late night power | | | • City gas<br>• District heating<br>• Kerosene | | | • Solar<br>• Geothermal<br>• Fuel cell | |

System type: boiler, gas/electric heat pump, cogeneration, district heating, absorption system, centrifugal chiller.

## 4. Development of a Non-Residential SPB Definition Framework

This section describes the SPB-type office building definition process, which utilizes the gathered data from the KEEI. This process is devised for any expert who has building stock survey data described in Section 3 to develop PBs. This is a unique contribution of this study to those countries that desire to develop PBs. Figure 3 shows the developed non-residential PB definition framework. Depending on the circumstances, some of the data described in this framework may not be obtained. In that case, it is possible to substitute the missing data using the best data available. The process has three main stages: data setup, determination of definition parameters, and verification of the results. Each stage is described in Sections 4.1–4.3.

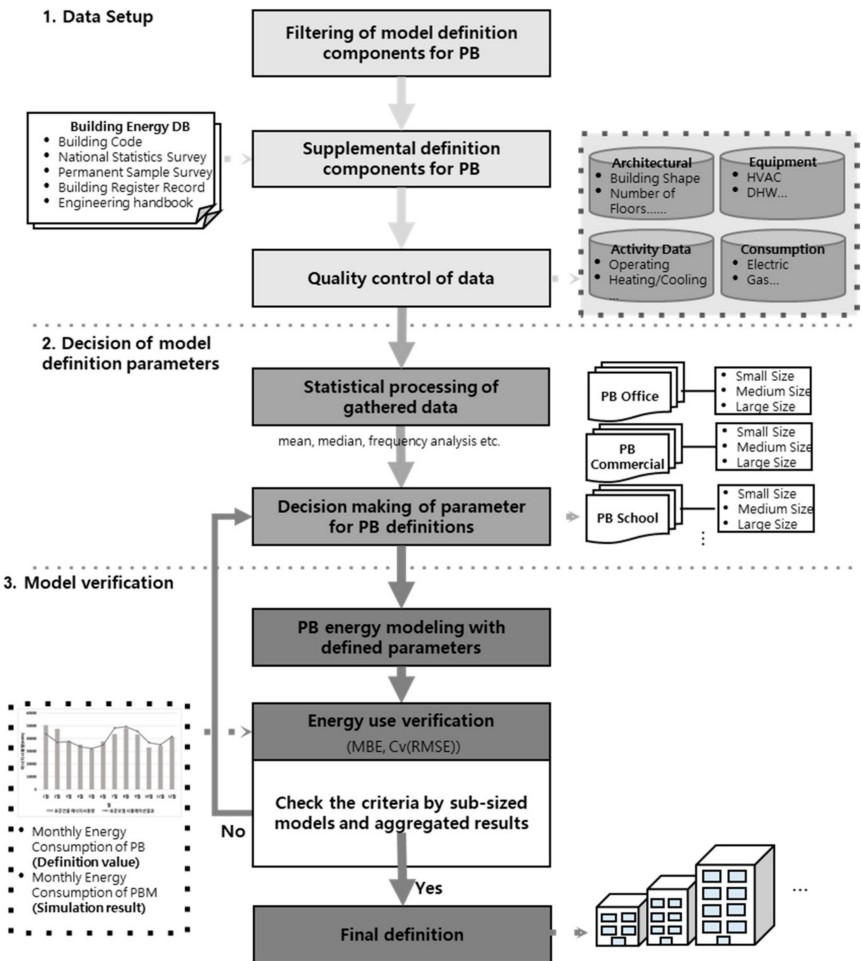

**Figure 3.** Suggested statistical analysis based prototypical building (SPB) type model definition framework.

### 4.1. Data Setup

In the data setup stage, the items necessary for the definition of the PB, which are presented in Table 3, were extracted from the responses. The items that contained insufficient data from the survey results were parameterized by engineering estimation; subsequently, quality control (QC) was performed. For example, the shape of the building, the materials of the main structure, and the building usage schedule can be directly determined through a questionnaire, but the thermal performance of the structure, which substantially influences energy consumption, is difficult to determine even by building management experts. In such cases, the construction year of the building can be used to determine the thermal performance by investigating the building design standards at the time of construction.

These types of data include the type of windows, thermal and optical performance, and airtight performance of the building.

After data preparation, QC is required. It is difficult to stipulate a set procedure for QC because of the unexpected errors in survey responses and investigator inputs. Therefore, it is necessary to examine the plausibility of outliers for each item and to consider correction and supplementation methods. However, basic QC procedures include missing data filling, outlier identification and removal, and a review of logical contradictions within the data. A representative metric for data QC is monthly energy consumption. If the monthly usage value was significantly higher or lower than the expected value range due to errors in units or inputs, it was excluded from the analysis.

### 4.2. Decision of Definition Parameters

Component values for PB definition are calculated by selecting the appropriate statistical indicators (average, median, frequency analysis) depending on the characteristics of the components. For example, for continuous variables, such as the building area, the appropriate measure is the sample average value. In the case of the building completion year, the mode is assumed to be the year with the greatest number of new buildings constructed rather than the average value. Table 4 shows the examples of the calculation methods for general architectural and lighting/equipment data. In the table, the variables are classified as categorical (CA) and continuous (CO) variables.

**Table 4.** Building general information variable statistics method

| Category of PB Definition | Variable Type | Statistics Method | |
| :---: | :---: | :---: | :---: |
| | | Mean | Mode |
| Typical floor shape | CA | | ● |
| Aspect ratio | CO | ● | |
| Building orientation | CA | | ● |
| Building area($m^2$) | CO | ● | |
| Condition: Unconditioned (%) | CO | ● | |
| Floor height(m) | CO | ● | |
| Year of completion | CO | | ● |
| WWR (%) | CO | ● | |
| Window type | CA | | ● |
| Glass type | CA | | ● |
| Structure of envelope | CA | | ● |
| Parking lot area | CO | ● | |
| Illumination density($W/m^2$) | CO | ● | |
| Lighting usage time (day/week & hour/day) | CO | ● | |
| Lighting control type | CA | | ● |

As the statistical values are calculated independently for each component, it is necessary to examine whether contradictory results occur among components at the building level. Therefore, there may be cases for which the defined results are examined from an architectural and architectural engineering perspective. In addition, some statistical values are modified according to engineering judgment. Table 5 shows the definition sources for an office PB (Large-1) determined by this two-step decision process. In the table, insufficient components that use complementary sources are indicated by internet map (IM), building energy code (Code), building registration record (BR), green together for energy consumption (GT) [23], and engineering judgment (EJ).

**Table 5.** Summary of definition sources and result of large 1 office PB.

| Definition Component | | Statistical Index | Definition Source | Definition Value |
|---|---|---|---|---|
| Building use | | Mode | KEEI | Office building (Large 1) |
| Orientation | | Mode | KEEI | Southeast |
| Aspect ratio | | Mode | KEEI, IM | Rectangle (1.2:1) |
| Floor height/Ceiling height | | Average | KEEI | 3.8 m/2.9 m |
| Typical floor area | | Average | KEEI | 1568 m$^2$ |
| Condition area | | Average | KEEI | 18,291 m$^2$ |
| Total area | | Average | KEEI, BR | 18,291 m$^2$ |
| Parking lot type and area | | Average | KEEI | Underground 4758 m$^2$ |
| Number of floors | | Average | KEEI, BR | Above ground 10/Underground 2 |
| Completion year | | Mode | KEEI, BR | 1987 |
| Wall | U-value | - | Code | 0.58 W/m$^2$·K |
| Wall | Structure type | Mode | KEEI | Concrete |
| Roof | U-value | - | Code | 0.29 W/m$^2$·K |
| Roof | Structure type | Mode | KEEI | Concrete + weather coat |
| Floor | U-value | - | Code | 0.58W/m$^2$·K |
| Floor | type | Mode | KEEI | Underground parking lot |
| Window | U-value | - | Code | 3.36 W/m$^2$·K |
| Window | Wall ratio(E, W, S, N) | Mode | KEEI, IM | 37%, 40%, 43%, 40% |
| Window | Glass type | Mode | KEEI | Double clear |
| Light density | | Average | KEEI | 8.15 W/m$^2$ |
| Energy supply system | | Mode Average | KEEI, EJ | Direct-fired absorption chiller: 909USrt/Turbo chiller:374USrt |
| Hot water system | | Mode Average | KEEI | Steam boiler (573 ton) |
| HVAC system | | Mode Average | KEEI, EJ | Constant air volume +fan coil unit (supply fan:13.3 kW, exhaust fan:9.4 kW) |
| Heating & cooling set point | | Average | KEEI | 21 °C, 26 °C |
| Energy supply | | Mode | KEEI | 8 month/yr, 10 Hr/Day |
| Hot water system | | Mode | KEEI | 7 month/yr, 4 Hr/Day |
| HVAC | Cooling | Mode | KEEI | May–Sep. 9:00–18:00 |
| HVAC | Heating | Mode | KEEI | Nov.–Mar. 9:00–18:00 |
| HVAC | Ventilation | Mode | KEEI | Jan.–Dec. 8:00–18:00 |
| Energy usage | Monthly electricity | Average | KEEI, GT | Figure 4a |
| Energy usage | Monthly fuel | Average | KEEI, GT | Figure 4b |

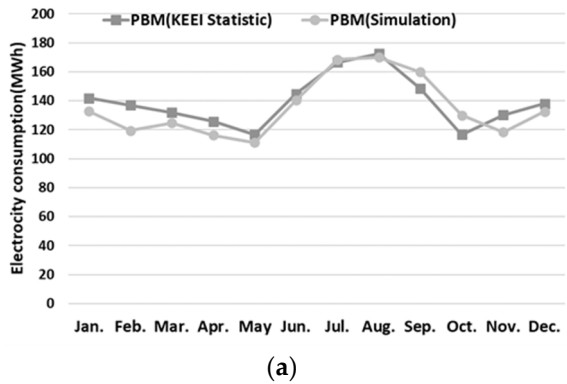 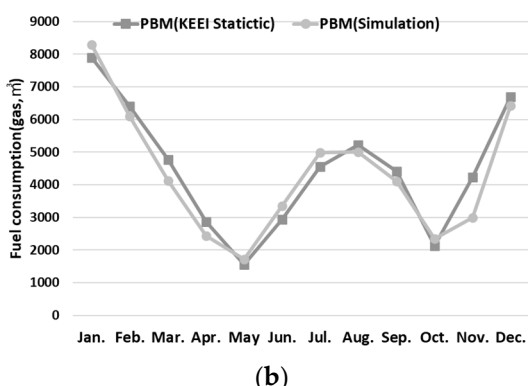

(**a**)  (**b**)

**Figure 4.** Comparison of monthly energy consumption of KEEI DB and PB office (Large-1) energy model: (**a**) Electricity; (**b**) Fuel.

### 4.3. Model Verification

One of the important reasons for developing a PB is to provide input into the building energy models and use the simulation results as objective data for prediction and verification. This process is used for developing energy-related technology and policies for the buildings. Therefore, it is important to verify that the PB energy model results reflect the energy consumption characteristics of the PB.

The results defined in Section 4.2 provide the energy model to evaluate the agreement between the simulation results and the actual energy consumption of the PB. If the model fails to satisfy the criteria, the final PB model calibration process is repeated. In general, the PB of a specific building stock requires sub-sized PB models based on the gross area to better reflect the energy consumption characteristics. Therefore, verification is also performed for each sub-sized PB model prior to the building stock-level verification process.

As verification criteria, the mean bias error (MBE) and the coefficient of variation of the root mean squared error (Cv(RMSE)) presented in ASHRAE guideline 14 [24] were utilized. These are the most commonly used verification criteria in building energy simulations. MBE is the average deviation between real and simulated data. A positive MBE implies overprediction. If the MBE is within ± 5%, the result is considered to have high reliability. RMSE is the root mean squared error between the actual and simulated data, i.e., the standard deviation between the two data sets. Cv(RMSE) is the rate of deviation between the actual data and simulation data and represents the uncertainty of the predictions. A monthly Cv(RMSE) value of 15% or less indicates high reliability. If any one of the indices cannot meet the criteria, the model should be calibrated by revising some of the defined values.

## 5. Case Study: Prototypical Office Building Energy Model Development, Verification, and Application

This section presents the results of the PB model definition, PB energy modeling, and verification. Subsequently, energy policy evaluation was performed. Five sub-sized PB definition models were constructed using the data extracted from the KEEI survey and the framework presented in Figure 3. One of the defined PB models was selected as the building energy model and for energy consumption verification. Finally, a case study that uses this energy model to evaluate the building energy policy was conducted.

### 5.1. Definition of PB Office Models

The definitions in Table 5 are difficult to use as input values in the building energy model simulator. Accordingly, they were reconstructed as shown in Table 6 as input values that can be simulated. Table 6 shows the definition results of prototypical office buildings defined according to the process described in Sections 3 and 4. To define the value of each "definition component," KEEI survey data were used, and the "Code" stands for Korea's Energy Saving Design Standards for Buildings. The gross area of buildings is a significant

factor influencing the early-stage design direction because of the building code and general design practices. As a result, building size has a fundamental influence on the energy consumption characteristics of the buildings. Therefore, the prototypical office buildings were divided into five groups by size. As the size of the building increases, many important definition components show changes, such as the window-to-wall area ratio (WWR) and the type of the HVAC system.

**Table 6.** Definition result of office PB models by size.

| | Definition Component | Mid-1 | Mid-2 | Large-1 | Large-2 | Large-3 |
|---|---|---|---|---|---|---|
| | Weather data | Seoul TMY2 | Seoul TMY2 | Seoul TMY2 | Seoul TMY2 | Seoul TMY2 |
| | Number of floors | 6 Floor, B2Floor | 9 Floor, B2Floor | 12 Floor, B4Floor | 16 Floor, B5Floor | 16 Floor, B2Floor |
| | Typical floor area | 600 m$^2$ | 720 m$^2$ | 1568 m$^2$ | 1752 m$^2$ | 5518 m$^2$ |
| | Condition Area | 3675 m$^2$ | 6180 m$^2$ | 18,291 m$^2$ | 26,886 m$^2$ | 77,238 m$^2$ |
| | Gross area | 4800 m$^2$ | 7923 m$^2$ | 25,087 m$^2$ | 36,795 m$^2$ | 99,332 m$^2$ |
| | Floor height | 3.6 m | 3.8 m | 3.8 m | 3.7 m | 4 m |
| | Ceiling height | 2.8 m | 2.9 m | 2.8 m | 2.8 m | 2.9 m |
| | Aspect ratio | 1.3:1 | 1.6:1 | 1.2: 1 | 1.2:1 | 1.8:1 |
| | Completion | | | 1987 | | |
| | Orientation | South | East | Southeast | East | East |
| Wall | U-value/Type | | | 0.58 W/m$^2$·K/ferroconcrete | | |
| Roof | U-value/Type | | | 0.29 W/m$^2$·K/Concrete + weather coat | | |
| Floor | U-value/Type | | | 0.58 W/m$^2$·K/Underground parking lot | | |
| Window | U-value/Type | | | 3.36 W/m$^2$·K/Fixed window with 0.6 SHGC | | |
| | WWR (E, W, S, N) | 33%, 35%, 39%, 37% | 43%, 47%, 46%, 46% | 37%, 40%, 43%, 40% | 41%, 43%, 46%, 45% | 44%, 44%, 49%, 47% |
| | Glass type | Double color | Double clear | Double clear | Double clear | Double color |
| | Light density | 5.17 W/m$^2$ | 6.03 W/m$^2$ | 8.15 W/m$^2$ | 5.49 W/m$^2$ | 6.35 W/m$^2$ |
| | Hot water system | Hot water (43 ton) | Steam boiler (115 ton) | Steam boiler (573 ton) | Steam boiler (683 ton) | Steam boiler (2342 ton) |
| Chilled water system | Direct-fired absorption | 155 USRT | 339 USRT | 909 USRT | 1139 USRT | 1907 USRT |
| | Turbo chiller | - | - | 374 USRT | 653 USRT | 1270 USRT |
| HVAC | Type | - | - | Constant air volume +Fan coil unit | | |
| | Supply fan/Exhaust fan | | | 13.3 kW/ 9.4 kW | 16.1 kW/ 14.7 kW | 12.7 kW/ 9 kW |
| | Cooling schedule | - | - | Jun–Sep 9:00~18:00 | Jun–Sep 8:00~18:00 | May–Sep 8:00~18:00 |
| | Heating schedule | - | - | Nov–Mar 9:00~18:00 | Nov–Mar 8:00~18:00 | Nov–Mar 9:00~18:00 |
| | Ventilation schedule | - | - | Jan–Dec 8:00~18:00 | Jan–Dec 8:00~18:00 | Jan–Dec 9:00~18:00 |
| | Hot water schedule | 6 month 6 Hr | 4 month 6 Hr | 7 month 4 Hr | 8 month 8 Hr | 6 month 10 Hr |
| | Heating/cooling setpoint | 21 °C, 26 °C | 21 °C, 26 °C | 21 °C, 26 °C | 21 °C, 26 °C | 21 °C, 26 °C |

### 5.2. PB Office (Large 1) Energy Model and Verification of Monthly Energy Consumption

The detailed building energy model of the PB office in Table 6 was constructed using eQUEST, a complete building energy simulation tool. The weather data used in the model was the Typical Meteorological Year (TMY) data of Seoul developed by the Architectural Environment and Energy Research Laboratory at Chungbuk National University. Modeling and verification were performed for prototypical office buildings of any size, but the verification results were described as representatives of Large-1 (Table 6).

Figure 4 shows a comparison of the monthly energy consumption from the KEEI survey and eQUEST-based PB office (Large-1) energy model simulation. The MBE and Cv (RMSE) values in Table 7 show the discrepancies in monthly electricity and gas consumption, but the verification criteria of the energy model were satisfied. The building energy model constructed using the definition of Large-1 prototypical office buildings represents the overall energy consumption characteristics of the office building stock.

**Table 7.** PB office (Large-1) energy model verification results with surveyed monthly energy use.

| Parameter | Electricity | Fuel |
| :---: | :---: | :---: |
| MBE | −2.76% | −3.38% |
| RMSE | 6.78% | 11.21% |

### 5.3. Case Study for Energy Policy Development with the PB

This case study used the PB office energy model defined in Section 5.2. Because the WWR of the prototypical office buildings defined in this study is greater than 40% on average, the heat loss and heat gain through the windows are expected to be significant. Accordingly, if there is a satisfactory design guide for selecting windows applied to new and existing office buildings, the construction of buildings with high energy efficiency is possible. This section proposes a simple design guide by analyzing the changes in building energy consumption with respect to various solar heat gain coefficients (SHGCs) based on the prototypical office building (Large-1) energy model. The prototypical office building energy model represents large office buildings. To normalize the value of different energy sources, the model is analyzed based on the primary energy consumption by end-use (cooling, heating, ventilation, lighting, hot water, and electricity) per unit area of the building. The SHGC applied to the PB office (Large-1) was 0.6, and the energy variations of the model with respect to the SHGC values compared to the PB model were analyzed.

Table 8 shows the changes in primary energy consumption per unit area by end-use with respect to SHGC (0.2–0.6). A comparison of the energy consumption of the prototypical office building models (SHGC = 0.6) shows that the decrease in the SHGC implies an increase in the total energy savings. A detailed analysis of each end-use indicates that the decreased solar heat gained through the window decreases the cooling energy. In addition, heating energy increases during the heating period. However, an overall energy savings of 3.4% was possible because the amount of ventilation energy savings was larger than the cooling reduction. This is expected because the amount of indoor heat gained by lighting, occupants, and equipment in large office buildings is usually larger than that of the heating load. More variables need to be considered to create an actual guide, but if the guide is written with only the SHGC for a large office building with a WWR of 40% or more, the SHGC should be designed to be as low as possible.

**Table 8.** Primary energy consumption and saving percentage by end-use per case.

| kWh/m²·yr | Base Line (SHGC 0.6) | Case 1 (SHGC 0.5) | Case 2 (SHGC 0.4) | Case 3 (SHGC 0.3) | Case 4 (SHGC 0.2) |
|---|---|---|---|---|---|
| Cooling | 48.9 | 45.9 (6.1%) | 43.6 (10.8%) | 41.6 (15.1%) | 40.2 (17.9%) |
| Heating | 27.4 | 30.8 (−12.4%) | 32.3 (−17.7%) | 35.3 (−28.9%) | 35.8 (−30.6%) |
| Ventilation | 63.7 | 60.1 (5.6%) | 57.2 (10.2%) | 55.1 (13.6%) | 58.1 (16.7%) |
| Lighting | 78.6 | 78.6 (0%) | 78.6 (0%) | 78.6 (0%) | 78.6 (0%) |
| Hot water | 16.2 | 16.2 (0%) | 16.2 (0%) | 16.2 (0%) | 16.2 (0%) |
| Plug | 92.8 | 92.8 (0%) | 92.8 (0%) | 92.8 (0%) | 92.8 (0%) |
| Total | 327.7 | 324.5 (1.0%) | 320.7 (2.1%) | 319.6 (2.5%) | 316.7 (3.4%) |

## 6. Conclusions

The PB model, which represents the energy use characteristics of building stock, can be very useful for building energy policy evaluation as well as technology development and evaluation. This study describes the process of conducting a large-scale building survey, analyzing the survey results, and applying engineering knowledge to develop prototypical office building models in Korea. Through this process, this study contributes to the energy policy field as follows:

(1) Survey components for a questionnaire on the energy consumption of office buildings were identified.

(2) Classification criteria of a PB with respect to input data as described in Figure 2 were developed.

(3) A framework was developed to collect information from building stocks, select PB definition components, determine component values, incorporate inputs into the energy model, and verify the representativeness of the model by observed energy consumption.

(4) The building energy policy development or evaluation process was demonstrated using the verified PB model. This case was made available as a reference to the experts who need to make informed decisions for policymaking and technology development and evaluation.

To update the values of the prototypical office buildings defined in this study, a survey on the energy consumption of each building should be performed. If the information is continually updated, it can be utilized in various fields such as tracing annual variations in building energy use and evaluation of energy policies, which correspond with our future research topics.

**Author Contributions:** Conceptualization, D.S.; methodology, D.S.; validation, H.-J.K.; formal analysis, H.-J.K.; investigation, D.-Y.C.; resources, D.-Y.C.; data curation, D.-Y.C.; writing—original draft preparation, H.-J.K.; writing—review and editing, D.S.; supervision, D.-Y.C.; project administration, D.-Y.C.; funding acquisition, D.-Y.C. All authors have read and agreed to the published version of the manuscript.

**Funding:** This work is supported by the Korea Agency for Infrastructure Technology Advancement (KAIA) grant funded by the Ministry of Land, Infrastructure and Transport (Grant 20AUDP-B151640-02).

**Conflicts of Interest:** The authors declare no conflict of interest.

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
