# Peer review of "Development and Verification of Prototypical Office Buildings Models Using the National Building Energy Consumption Survey in Korea"

_sustainability, doi:10.3390/su13073611_

Round 1
Reviewer 1 Report
The content of the article is of a very high level. In my subjective opinion, it is suitable for publication without making changes to the content. Editing changes are recommended, but they do not apply to comments from the reviewer.
Author Response
Thank you very much for your opinion. As you indicated, some editorial errors are revised.
Reviewer 2 Report
The comments are below:
Presentation needs improvement. For Figure 1, the x axis is confusing with the numbers. Please consider adding an angle to the year so it becomes clearer. Also explicitly define what is the U-value.
Correct “Chapter XX” to “Section XX”
Define all acronyms such as DHW, AHU etc. Also reduce the number of acronyms used possible to enhance readability. Generally, if acronyms are used less than three times then can be removed.
In Table II, correct “II. energy facilities”.
In line 396, correct “addtion"
Please clearly state the contribution to knowledge in the Introduction by using bullet points.
The literature review can be improved. For example, novel methods for residential building energy management has been discussed:
- Two-stage optimal scheduling of air conditioning resources with high photovoltaic penetrations. Journal of Cleaner Production, 241, p.118407.
- A Time-Synchronized ZigBee Building Network for Smart Water Management. In Smart Grids and Big Data Analytics for Smart Cities(pp. 307-343). Springer, Cham.
In Table 1, the energy consumption for renewable energy is not clear. How is this quantified? Should this be part of electricity consumption?
“The MBE is a criterion for determining the bias of energy prediction, and it is considered to have high reliability when the MBE result is within ± 5%. Cv (RMSE) represents the degree of diffusion between real data and simulation data. The monthly Cv (RMSE) indicates that the reliability is high when it is 15% or less. If any one of the indexes cannot meet the criteria, the model should be calibrated by revising some defined values.” These statements need more elaboration with the numbers define. Why are these the verification criteria? Are there other metrics? Some reference can be used to support the statements. Please note MBE and RMSE are used for many applications and the acceptable margin is application dependent such as electricity forecasting: 2020. Multi-view Neural Network Ensemble for Short and Mid-term Load Forecasting. IEEE Transactions on Power Systems. DOI: 10.1109/TPWRS.2020.3042389
The modelling is not clear in particular the data setup. For example, Which Engineering handbook, building register record are used for the case study?
Author Response
Thank you very much for your valuable comments. Please find the attached the responses of the authors.

Reviewer 3 Report
The submitted article is recommended for publication after minor revision.
The detailed comments and suggestions are contained in the attached file.

Author Response
Thank you very much for your valuable comments. Please find the attached responses of the authors for your comments.

Round 2
Reviewer 2 Report
The authors have made an improvement with the paper. However, English is still a major concern and the present version is still difficult to read.
A proper proofreading is a must. For example, correct “after the two decades” should be: “after the last two decades” or “after two decades”.
“framework development is implemented” should be: a “framework is implemented” or “framework is developed and implemented”
“Statistical analysis based on a prototypical building (SPB) which uses statistical data that are regularly published by state and public institutions for definition.” Rewrite as “Statistical data are regularly defined by state and public institutions with statistical analysis based on prototypical building (SPB).”
Author Response
Thank you very much for the second review.
At the time of the first submission, this paper had already been proofread by a native speaker. However, I don't think the proofreading was perfect, and the author's review wasn't perfect.
Therefore, re-proofreading was done again by native speakers and nearly 1000 corrections were completed. Since it is difficult to summarize the revised items one by one, the tracking-activated paper was attached, and the proofreading company's work completion document was also attached.
Thank you for reviewing to make this paper better.

Round 3
Reviewer 2 Report
The authors have polished the English and the reviewer believes that the manuscript meets the standard of the journal.